# PTPRK, an EGFR Phosphatase, Is Decreased in CeD Biopsies and Intestinal Organoids

**DOI:** 10.3390/cells12010115

**Published:** 2022-12-28

**Authors:** Merlin Nanayakkara, Claudia Bellomo, Francesca Furone, Mariantonia Maglio, Antonella Marano, Giuliana Lania, Monia Porpora, Martina Nicoletti, Salvatore Auricchio, Maria Vittoria Barone

**Affiliations:** 1Department of Translational Medical Science, Section of Pediatrics, University Federico II, Via S. Pansini 5, 80131 Naples, Italy; 2ELFID (European Laboratory for the Investigation of Food Induced Diseases), University Federico II, Via S. Pansini 5, 80131 Naples, Italy

**Keywords:** PTPRK, EGFR, celiac disease, intestinal organoids

## Abstract

Background & Aims: Celiac disease (CeD) is an immune-mediated enteropathy triggered in genetically susceptible (HLA-DQ2/8) individuals by a group of wheat proteins and related prolamins from cereals. The celiac intestine is characterized by an inversion of the differentiation/proliferation program of the enterocytes, with an increase in the proliferative compartment and crypt hyperplasia, which are the mechanisms that regulate the increased proliferation in CeD that arenot completely understood.The aim of this study is to understand the role of Protein Tyrosine Phosphatase Receptor Type K (PTPRK), a nodal phosphatase that regulates EGFR activation in the proliferation of the enterocytes from CeD biopsies and organoids. Methods: The levels of PTPRK were evaluated by RT PCR, western blot (WB) and immunofluorescence techniques in intestinal biopsies and organoids from CeD patients and controls. Additionally, pEGFR and pERK were evaluated by WB and proliferation by BrdU incorporation. PTPRK si-RNA was silenced in CTR organoids and was overexpressed in CeD organoids. Results: PTPRK was reduced in Gluten Containing Diet–Celiac Disease (GCD–CeD) and Potential-Celiac Disease(Pot-CeD) biopsies (*p* < 0.01–*p* < 0.05) whereas pEGFR (*p* < 0.01 *p* < 0.01), pERK (*p* < 0.01 *p* < 0.01) and proliferation were increased. (*p* < 0.05 *p* < 0.05) respect to the controls.The CeD organoids reproduced these same alterations. Silencing of PTPRK in CTR organoids increased pEGFR, pERK and proliferation. The overexpression of PTPRK in CeD organoids reduced pEGFR, pERK and proliferation. Conclusions: modulation of PTPRK levels can reduce or increase pEGFR, pERK and proliferation in CeD or CTR organoids, respectively. The CeD organoids can be a good model to study the mechanisms of the disease.

## 1. Introduction

Celiac disease (CeD) is an immune-mediated enteropathy triggered in genetically susceptible (HLA-DQ2/8) individuals by a group of wheat proteins and related prolamins from cereals [1]. In particular, these proteins damage the intestinal mucosa through the inflammation generated by the immune response that is both innate and adaptive against gliadin, causing crypt hyperplasia through uncontrolled proliferation of enterocytes [2,3,4,5].

The celiac intestine is characterized by reduced differentiation of the enterocytes, resulting to a reversal of the differentiation/proliferation program of the intestinal tissue. This may lead to complete villous atrophy and increased proliferation with cryptic hyperplasia [6,7].

In two patient populations, Gluten Containing Diet–Celiac Disease (GCD–CeD) and Gluten Free Diet–Celiac Disease (GFD–CeD), at the level of the duodenal biopsies, increased activity both of the EGFR/EGF (Epidermal Growth Factor Receptor/Epidermal Growth Factor) system and of the ERK (Extracellular Signal-Regulated Kinases) molecule has been demonstrated. Moreover, activation of EGFR/ERK pathway has been linked to the enhancement of enterocytes proliferation [5,7,8]. It is interesting to note that both gluten and gliadin peptide P31-43 can increase proliferation in GFD–CeD patients biopsies in vitro and after gluten challenge in vivo and that they are able to affect several other biological pathways in vitro in cells [8,9,10,11,12,13] and also in animal models [14,15].HLA-DQ 2/8 isotypes have been strongly associated with CeD, this genetical predisposition is necessary but not sufficient to the insurgence of the disease. Repeated GWAS (Genome Wide Association Study) analysis led to the identification of a total of 39 non-HLA sensitivity loci that explain only 10–14% of the CeD heritability, these in combination with the HLA locus could explain 50% of the CeD heritability [16,17].In order to prioritise and annotate functional SNPs (Single Nucleotide Polymorphisms), pathways and genes affected in CeD, Kumar et al. have applied integrative approaches [17]. Results from 1469 blood samples were combined with co-expression analyses to give priority to causative genes, from different blood immune cell populations. Data from pathways and tissue-specific expression analyses on these genes identified a role in CeD pathogenesis for Lipoma-Preferred Partner (LPP) [18,19,20], C1ORF106 (C1 Orfan 106), ARGHAP31 [21] and Protein Tyrosine Phosphatase Receptor Type K (PTPRK) genes, which play a role in actin-cytoskeleton rearrangement, cell-cell adhesion and in the Epidermal Growth factor (EGF)/EGF Receptor (EGFR) pathway activation [17,22]. PTPRK, an EGFR phosphatase, is reduced in CeD biopsies respect to controls [17,23].

Proliferation, metabolism, differentiation, motility, cytoskeletal organization, cell-cell interactions, development and other cellular processes are under control of reversible protein phosphorylation regulating signal transduction pathways [24]. In particular, both Protein-Tyrosine Kinases (PTKs) and Protein-Tyrosine-Phosphatases (PTPs), two classes of counteracting enzymes, control the level of protein tyrosine phosphorylation. New evidence indicates that, depending on the specific pathway, protein tyrosine dephosphorylation may be of equal or greater importance to the regulation of cell function than protein tyrosine phosphorylation [25]. The EGFR is a component of Receptor Protein-Tyrosine Kinase (RPTK) superfamily. When a ligand binds to the N-terminal extracellular domain of EGFR, it stabilises homodimerization and heterodimerisation with other members of EGFR family and promotes trans tyrosine phosphorylation of the intracellular C-terminal domain. It has been demonstrated that the aberrant regulation of EGFR promotes multiple tumorigenic processes by stimulating angiogenesis, proliferation and metastases. EGFR and/or its ligands have a significant role in many different types of solid tumours and are most common in human epithelial cancers. The main mechanism by which EGFR modulates several transduction pathways is the phosphorylation of its tyrosine and this particular step must be strictly regulated. PTP-catalysed dephosphorylation of EGFR is one of these mechanisms for regulation. PTPRK has been demonstrated to dephosphorylate EGFR; in CHO (Chinese Hamster Ovary) cells that lack PTPRK, there is increased proliferation [26]. Aim of this study is to analyse and understand the role of PTPRK, an EGFR phosphatase, in the CeD intestinal epithelium from biopsies and organoids.

## 2. Materials and Methods

### 2.1. Biopsies

For organ culture studies, duodenalbiopsies were obtained from patients at GCD-CeD (20 patients, mean age of 10 years), controls affected by gastroesophageal reflux (20 subjects, mean age of 13 years) and Pot-CeD patients (8 subjects, mean age of 11 years). The age range for all of them was 2–16 years. The group of patients with GCD-CeD (Marsh T3c) and the Pot-CeD group had positive serology (anti-tTg antibodies >50 U/mL and in a range between 9 and 60 U/mL and positive EMA, respectively) (Table 1). Eurospital kit EU-tTG(cat. 9105, Trieste, Italy). was used to evaluate anti-tTg antibody titre. Specimens were harvested, snap-frozen in liquid nitrogen, embedded in OCT (cat. 05-9801, Bio Optica, Milan, Italy) and stored at −80 °C.

### 2.2. Organoids

Two or more duodenal biopsies from GCD-CeD patients and controls were taken during routine gastro-duodenoscopy (Table 2) and placed in 10 mL of isolation buffer with the addition of 2 Mm EDTA (cat.15575020, Thermo Fisher, Milan, Italy) and 0.5 Mm DDT (D0632, Sigma-Aldrich, Milan, Italy), as described previously [27]. After 1 h of agitation at 4 °C, the biopsy samples were washed with the isolation buffer, filtered through a 70 µm strainer (Falcon, NY, USA) and centrifugated at 500× *g* for 5 min, then the crypts were resuspended in 40 µL of ice-cold Matrigel matrix (cat.35623, Corning, Milan, Italy) to permit three-dimensional growth in 48 well plates. Afterwards, 300 µL cell culture medium enriched with supplements (CM-S) was added to each well [27]. To pass from a 3D to a 2D structure, organoids were openly seeded on 35 mm multiwellspre-treated with Matrigel (1:40 in PBS). We generally processed the 2D organoids within 24–72 h after seeding.

### 2.3. Immunofluorescence Staining of Biopsies

PTPRK protein levels and localisation were assessed by immunofluorescence. An amount of 5 µm cryostat sections from each biopsy were treated with acetone (10 min), Pot–CeD patients, controls and blocked with 1% bovine serum albumin (BSA) in PBS for 15 min at room temperature (RT) and stained with rabbit anti-PTPRK (cat. ab222249, Abcam, Microtech, Naples, Italy) antibody for 1h at RT.Then, the sections were probed with Alexa 488 anti-rabbit (1:50; cat.11001, Milan, Italy) for 45 min at room temperature and counterstained with Hoechst (cat. 23491454, Sigma-Aldrich, Milan, Italy), mounted with Mowiol (cat. 9002895, Sigma-Aldrich, Milan, Italy) and observed under a Zeiss LSM510 or Zeiss LSM700 confocal microscopes (laser scanning microscope) (Germany). Images were obtained with a 63× objective unless differently stated [10].

### 2.4. BrdU Cell Proliferation Assay

PTPRK was silenced, or not, in CTR organoids and was overexpressed, or not, in GCD CeD organoids seeded on 2D on 15 μ-Slid Angiogenesis (cat.81506, ibidi Milan, Italy) coated with Matrigel diluted 1:40 in PBS for 2 d.

BrdU (cat. B5002, Milan, Italy), a marker of S phase, was added in all samples for 18 h before fixation and immunofluorescence staining. BrdU was detected with a monoclonal antibody (RPN20AB, GE Healthcare, Buckinghamshire, UK) and with an anti-mouse-Alexa-633 (1:10, cat. A21063, Milan, Italy) conjugated as secondary antibody. Nuclei were highlighted by Hoechst staining (Sigma-Aldrich, Milan, Italy). Cell proliferation was assessed by BrdU incorporation assay as described elsewhere [10]. LSM 510 Zeiss microscope was used to acquire and analyse the images.

### 2.5. Western Blot

Biopsy fragments from duodenum obtained from GCD–CeD and organoids, were processed [10,28] After removing matrigel by Cell Recovery Solution (cat. 354270, Corning Milan Italy), organoids were homogenized in 50 µL tissue homogenization buffer [28]. The cell lysates (10 μg/mL) were analysedby SDS-PAGE and transferred onto nitrocellulose membranes by Trans Blot Turbo (cat.1704158, BioRad, Milan, Italy). The membranes were blocked with 5% non-fat dry milk and probed with anti-PTPRk (cat. ab222249, Abcam, Microtech, Naples, Italy) and rabbit anti-pY-ERK1/2, (cat.20869, Elabscience, Microtech, Naples, Italy), anti-P-EGFR (Y1068) (cat.3777,Cell Signalling, Euroclone Milan, Italy) and anti-EGFR (cat.2232, Cell Signalling, Euroclone Milan, Italy), mouse anti-GAPDH, (cat. G8795, Sigma-Aldrich, Milan, Italy) and with rabbit anti-ERK1/2 (cat.31374, Elabscience, Microtech, Naples, Italy) were used as loading controls for biopsies and organoids, respectively. Then, two 10 min exposure using ECL (cat. RPN2209, GE Healthcare, Amersham, Buckinghamshire, UK) allowed to visualise the bands of interest. The bands intensity was evaluated by integrating all the pixels of the band after subtraction of the background to calculate the average of the pixels surrounding the band.

### 2.6. Crypt Epithelial Cell Proliferation Test

A total of 4µm cryostat sections duodenal biopsies from 4 GCD-CeD, 4 Pot-CeD and 4 CTR patients were used for evaluating crypt epithelial cell proliferation by immunohistochemistry using Ki67 antigen detection (Table 3). The sections, after a pre-incubation of 10 min with rabbit normal serum (1:100; X0902,Dako, Milan, Italy), were probed with the primary mouse monoclonal antibody Ki67 (1:200; M7240, Dako, Milan, Italy) for 1h and the secondary antibody with rabbit anti-mouse (1:25; Z0259, Dako, Milan, Italy) for 30 min followed by a 30min step with alkaline phosphatase and monoclonal mouse anti-alkaline phosphatase immuno-complexes (mouse APAAP 1:40; K0670, Dako, Milan, Italy). Lastly, the sections were incubated with New Fuchsin for few minutes. A light microscope Axioskop2 plus (Zeiss, Germany) was used to observe sections. Each sample shown more than 500 epithelial cells and the number of Ki67-positive cells was expressed as a percentage of the total number of positive crypt epithelial cells.

### 2.7. mRNA Analysis

Total mRNA was extracted from biopsies of patients with GCD–CeD, Pot–CeDand controls using TRIZOL reagent (cat.10296028, Ambion^®^-Life Technologies Milan, Italy). The mRNA concentration was measured using a Nanodrop^®^ spectrophotometer (Thermo Fisher, Milan, Italy), the RNA quality was analysed using agarose gel electrophoresis in Tris/Borate/EDTA buffer (TBE, Sigma, Milan, Italy). The RNA (1 µg) was reverse transcribed into cDNAs. The gene expression assay used for PTPRK gene was Hs00267788_m1 (Cat#4331182, Thermo Fisher Scientific).Real-time PCR were performed with approximately 40 ng of cDNA templates, according to the manufacturer’s protocol (TaqMan^®^ Gene Expression Assay (cat.4369016, Thermo Fisher, Milan, Italy) [8].

### 2.8. PTPRK Silencing

PTPRK mRNA (Hs PTPRK 1) was used for silencing experiments. To evaluate transfection efficiency was employed non-specific siRNA (Cat. 1022564) (MAPK1), scrambled mRNA and scrambled mRNA sequences (Cat.1027284) (All Stars Negative). HIPerFect Transfection Reagent (cat.301705) was used for transfections. All these products were purchased from QIAGEN, Milan, Italy. The CTR organoids were seeded on 35 mm multiwells (Corning, Milan, Italy) for 24 h. In sum, the cells grown in cell culture medium supplemented with CM-S were added with a transfection mix composed by 6 μg of siRNA, diluted in 100 micro-litres of culture medium without serum to give a final siRNA concentration of 50 nM and 20 µL of HIPerFect Transfection reagent. The transfection mix was vortexed and added drop wise onto the cells that were incubated for 72 h. The cells were then processed for Western blot (WB) analysis and proliferation assay [8,10].

### 2.9. PTPRK Overexpression

Following the manufacturer’s instructions PTPRK transfection was conducted with TT210003 MegaTran2.0 Transfection Reagent (OriGene Clone, Milan, Italy). In sum, the human spheroids from GCD–CeD were incubated in cell culture medium supplemented with CM–S for 24 h. The transfection mix was done by mixing MegaTran 2.0 and DNA in a 3:1 ratio in 100 micro-litres of serum free DMEM. The day after, the transfection mix was incubated for 10–15 min at room temperature to allow transfection complexes to form and were then added to the cells that were incubated for 24 h. Cells were than processed for WB and proliferation assay.

### 2.10. Statistical Analysis

For statistical analyses and graphical representationswas used GraphPad Prism 6.lnk software (GraphPad Software, San Diego, CA, USA). The statistical analysis of the differences was performed using Student’s *t*-test and Mann–Whitney test A *p* value < 0.05 was considered statistically significant.

## 3. Results

### 3.1. In Enterocytes from GCD–CeD and Pot–CeD Biopsies, PTPRK Is Decreased, pEGFR, pERK and Proliferation Are Increased

PTPRK protein is decreased in the intestine of acute CeD patients at GCD with villus atrophy as shown by WB analysis (Figure 1A,B). In fact, PTPRK protein was decreased in the protein’s lysate from intestinal biopsies of GCD–CeD patients compared to controls. This observation was confirmed by immunofluorescence staining of the PTPRK proteins (Figure 1C,D),which showeda significant decrease of the fluorescence intensity of the PTPRK staining of intestinal biopsies from GCD–CeD both in the crypts and in the villi respect to controls. Moreover, the immunofluorescence staining revealed that PTPRK localization is mainly at the level of the intestinal epithelium of crypts and villi both in controls and in CeD biopsies (Figure 1C,D). Interestingly a similar decrease of the PTPRK protein and a similar localization in the epithelial cells was found in the biopsies from Pot–CeD with normal villi architecture at GCD (Figure 1A–D).

The decrease of PTPRK is generally linked to the increase inEGFR (pEGFR) and ERK (pERK) phosphorylation together with the increase inproliferation. For this reason, we have evaluatedGCD–CeD and Pot–CeD biopsies for pEGFR levels respect to CTR biopsies. In Figure 2A,B, it isshown by WB analysis that pEGFR is increased in GCD–CeD and Pot–CeD respect to CTR biopsies. As expected, pERK was also increased in GCD–CeD and Pot–CeD respect to CTR biopsies (Figure 2C,D). Moreover, in these same biopsies the proliferation marker Ki67 was increased in the intestinal crypts respect to controls as shown by immunohistochemistry experiments (Figure 2E,F and Table 4). The combination of these experiments show that, in GCD–CeD and Pot–CeD with low levels of PTPRK, the phosphorylated form of EGFR and ERK were both increased together with proliferation.

### 3.2. Intestinal Organoids from CeD Patients Reproduce the Increase inpEGFR, pERK and Proliferation Found in GCD–CeD and Pot–CeD Biopsies

Intestinal organoids derived from CeD patients at GCD were analysed for PTPRK mRNA levels by quantitative PCR analysis. In Figure 3, aReal Time PCR analysis of intestinal organoids derived from CeD patients have less PTPRK mRNA respect to intestinal organoids derived from controls. Additionally, PTPRK protein levels were also reduced in these organoids as shown in Figure 3B,C. Moreover, in these same organoids pEGFR (Figure 3B,D) and pERK (Figure 3B,D) were increased as shown by WB analysis.

Furthermore, BrdU incorporation, a marker of S phase entry of the cell cycle, was increased in CeD organoids nuclei respect to controls (Figure 3G,H) showing that CeD organoids enterocytes proliferate more respect to controls. Taken all together these data indicate that intestinal organoids from CeD patients reproduce the same increase inpEGFR, pERK and proliferation present in intestinal biopsies from GCD–CeD and Pot–CeD intestinal biopsies.

### 3.3. Silencing of PTPRK Protein in CTR Organoids Induced Increase in pEGFR, pERK and Proliferation

To demonstrate the correlation between the low levels of PTPRK and the increase inpERGF, pERK and proliferation found in CeD organoids, we decided to reduce the PTPRK levels in CTRs organoids by silencing PTPRK (Figure 4A,B). In CTR organoids with PTPRK silenced both pEGFR and pERK increased respect to the not silenced controls (Figure 4A,C–E). Moreover, in CTR organoids with siPTPRK, proliferation measured by BrdU incorporation was increased.

Showing that the reduction inthe levels of the PTPRK protein by silencing mRNA was able to modulate the levels of pEGFR, pERK and the proliferation in CTRs reproducing the same alterations found in CeD organoids.

### 3.4. Overexpression of PTPRK Protein in CeD Organoids Reduced pEGFR, pERK and Proliferation

Afterword, we have overexpressed PTPRK in CeD organoids to confirm that the levels of pEGFR, pERK and proliferation were dependent on PTPRK levels. In CeD organoids with PTPRK overexpression, both pEGFR and pERK decreased respect to the not transfected CeD organoids (Figure 5A,C–E). BrdU incorporation also decreased in CeD organoids overexpressing PTPRK. This shows that the increase inthe levels of the PTPRK protein by overexpression was able to modulate the levels of pERGF, pERK and the proliferation in CeD organoids.

## 4. Discussion

In this manuscript we have analysed the role of the PTPRK phosphatase on the increase inpEGFR, pERK and proliferation in CeD biopsies and intestinal organoids. In CeD biopsies, both GCD–CeD and Pot–CeD patients the levels of PTPRK that decreased in respect to controls, indicating that the reduction inthe PTPRK protein was independent from the intestinal atrophy. Moreover, in the biopsies with low levels of PTPRK there was an increase inpEGFR, pERK and crypts epithelial cells proliferation, indicating that also in intestinal epithelium, PTPRK can have a role in the regulation of the signalling from EGFR to proliferation. Subsequently, we showed that intestinal organoids derived from CeD biopsies had low levels of PTPRK, together with increased pEGFR, pERK and crypts enterocytes proliferation. Therefore, intestinal organoids from CeD patients reproduce the alterations of the EGFR/ERK/proliferation pathway found in CeD biopsies.

To show the causal link between the PTPRK levels and the state of activation of EGFR and ERK together with the increased number of proliferating enterocytes, we have silenced PTPRK in CTR organoids to create the same phenotype present in CeD organoids. In CTR organoids with reduced levels of PTPRK there was an increase inpEGFR and pERK together with increased proliferation. Indicating that the levels of PTPRK protein can regulate the EGFR pathway that leads to proliferation. On the other side, the overexpression of the PTPRK protein in CeD organoids was able to reduce both EGFR/ERK phosphorylation state and the cell proliferation.

In CeD, the role of the EGFR/ERK/proliferation pathway has been established [5,9]. In fact, in CeD biopsies and in fibroblasts, pEGFR and pERK are increased together with increased proliferation, that is EGFR and ERK dependent [7,8,11]. In CeD biopsies and fibroblasts EGFR is delayed in the early endocytic vesicles and stays longer activated respect to CTRs [8]. Moreover, gliadin and gliadin peptide P31–43, further delay the endocytic pathway and activate the same EGFR/ERK/proliferation pathway both in CeD biopsies and in control’s biopsies [7,11]. Interestingly, EGFR is also involved in the innate immune response mediated by IL15R-alpha upon gliadin treatment in CeD intestinal biopsies [28].

PTPRK is a nodal phosphate that regulates the signalling from tyrosine kinase receptors, including EGFR [26]. EGFR (ErbB1) belongs to the receptor protein-tyrosine kinase (RPTK) superfamily. Aberrant regulation of EGFR has been shown to promote multiple tumorigenic processes by stimulating proliferation, angiogenesis, and metastasis [29,30]. PTPRK has been found co-expressed with THEMIS in the celiac intestinal mucosa [23].

Moreover, other phosphatases of the PTP family are involved in the pathogenesis of inflammatory bowel diseases (IBD), where an association with the lack of PTP proteins has been linked to inflammation [30].

Intestinal organoids from CeD patients not only reproduced the alterations found in CeD biopsies, but also allowed challenge the intestinal epithelium by silencing and overexpressing PTPRK. In this way, by silencing PTPRK in CTR organoids, we were able to reproduce some of the mechanisms of the disease such as the increase inpEGFR/pERK and proliferation found in CeD organoids. On the other side overexpressing PTPRK in CeD organoids reduced to normal the levels of pEGFR, pERK and proliferation. In sum, we were able to recreate the CeD phenotype in CTRs organoids and to “cure” the CeD organoids modulating the PTPRK protein levels using as read outs pEGFR/pERK levels and proliferation. Additionally, the link between PTPRK and TGF-B is known as is the one between Notch and TGF-beta [31]. It is possible to hypostasize that they can have a role in the mechanisms of the villus atrophy, in which way it is too soonto established.

In conclusion, organoids turned out to be a good model to study some pathogenetic mechanisms of the disease, as they reproduced the alterations found in CeD biopsies. They can be a very simple model allowing to study the interaction of the intestinal epithelium and food or other environmental agents. This model can be extended to other diseases in which the intestinal epithelium plays a nodal role in the pathogenetic mechanism such as IBD and diabetes. On the other side CeD is a complex disease, in which several different factors and mechanisms can contribute to the pathogenetic events. Nevertheless, it is possible to increase the complexity of the intestinal epithelial organoids to finally reproduce the complete intestinal model and study the contribution of each intestinal cellular compounds to the mechanisms of the disease.

## Figures and Tables

**Figure 1 cells-12-00115-f001:**
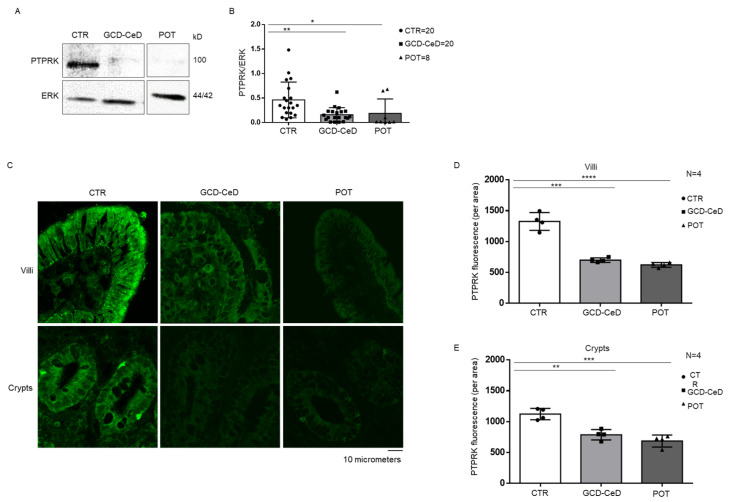
**PTPRK protein is decreased in intestinal enterocytes from CeD biopsies** (**A**). Western blot analysis of total protein lysates of intestinal biopsies from controls (CTR), CeD patients in the acute phase of the disease (GCD–CeD) and CeD potential patients (Pot–CeD). Upper lines were blotted with an antibody against PTPRK. Anti-ERK (extracellular-signal-regulated kinase) antibody was used as loading control (bottom lines). Representative images are shown. (**B**). Densitometric analysis of the PTPRK/ERK bands from CTR, GCD–CeD and Pot–CeD.Number of patients analysed are indicated (N = 5). Columns represent the mean and bars the standard deviation. Student’s t test: * = *p* < 0.05; ** = *p* < 0.01. (**C**). Immunofluorescence images of intestinal biopsies crypts and villi from CTR, GCD–CeD and Pot–CeD stained with anti-PTPRK antibodies. (**D**,**E**). Statistical analysis of PTPRK immunofluorescent staining in villi and crypts from CTR, GCD–CeD and Pot–CeD. Columns represent the mean and bars the standard deviation. Student’s t test: * *p* < 0.05; ** = *p* < 0.01; *** = *p* < 0.001; **** = *p* < 0.0001. N represent number of organoids evaluated.

**Figure 2 cells-12-00115-f002:**
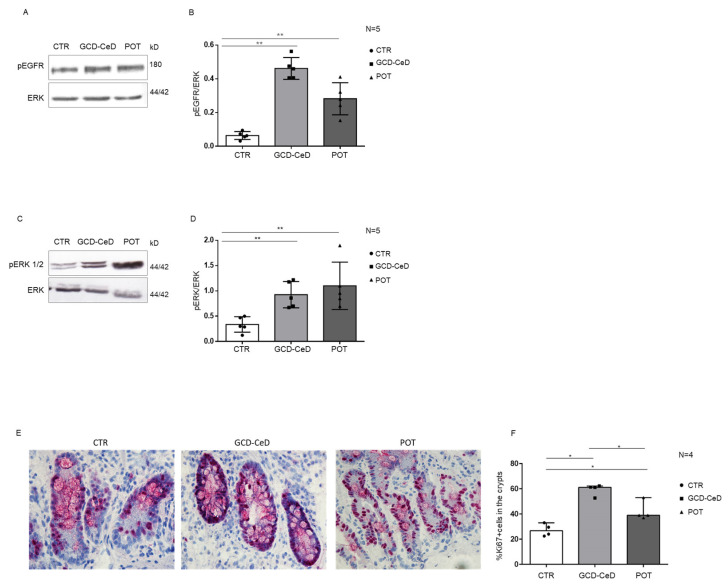
**pEGFR, pERK proteins and proliferation are increased in CeD biopsies with low levels of PTPRK.** (**A**). Western blot analysis of total protein lysates of intestinal biopsies from CTR, GCD–CeD and Pot–CeD. Upper lines were blotted with an antibody against pEGFR. Bottom lines were blotted with anti-ERK antibodies as loading control. Representative images were selected. (**B**). Densitometric analysis of the pEGFR/ERK bands from CTR, GCD–CeD and Pot–CeD.The numbers of patients analysed are indicated. Columns represent the mean and bars the standard deviation. Student’s *t* test: ** = *p* < 0.01. (**C**). Western blot analysis of total protein lysates of intestinal biopsies from CTR, GCD–CeD and Pot–CeD. Upper lines were blotted with an antibody against the phosphorylated from of ERK (pERK). Bottom lines were blotted with anti-ERK antibodies as loading control. Representative images were selected. (**D**). Densitometric analysis of the pERK/ERK bands from CTR, GCD–CeD and Pot–CeD.The numbers of patients analysed are indicated. Columns represent the mean and bars the standard deviation. Student’s *t* test: ** = *p* < 0.01. (**E**). Immunohistochemistry analysis of crypts from biopsies of CTR, GCD–CeD and Pot–CeD patients stained with antibodies against Ki67. Red dense spots indicate positive nuclei. Total nuclei are in blue. (**F**). Statistical analysis of the percentage of Ki67 positive nuclei in the crypts respect to total nuclei. Columns represent the median and bars the range. Mann–Whitney test. * = *p* < 0.05.

**Figure 3 cells-12-00115-f003:**
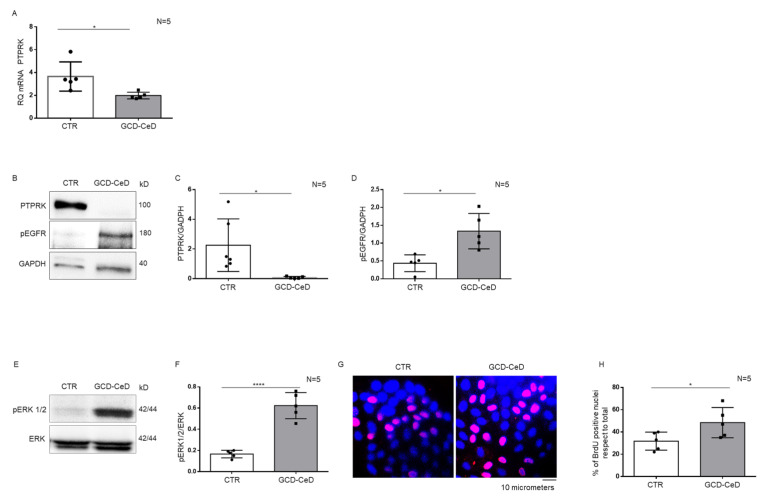
**CeD organoids have low levels of PTPRK mRNA, and protein, pEGFR, pERK and proliferation are increased**. (**A**). PTPRK mRNA levels in CTR and GCD–CeD organoids analysed by Real Time PCR. Columns represent the mean and bars the standard deviation. Student’s *t* test: * = *p* < 0.05. (**B**). Western blot analysis of total protein lysates of intestinal organoids from CTR and GCD–CeD. The membrane was blotted with antibodies against PTPRK, pEGFR and GAPDH as loading control. (**C**,**D**).Densitometric analysis of the PTPRK/GAPDH and pERK/GAPDH bands. Columns represent the mean and bars the standard deviation. Student’s *t* test: * = *p* < 0.05. (**E**). Upper lines were blotted with an antibody against pERK. Bottom lines were blotted with anti-ERK antibodies as loading control. (**F**). Densitometric analysis of the pERK/ERK bands. Columns represent the mean and bars the standard deviation. Student’s *t* test: **** = *p* < 0.0001. (**G**). Immunofluorescence images of anti-BrdU staining on intestinal organoids from CTR and GCD–CeD patients. Red nuclei are positive for BrdU staining. (**H**). Statistical analysis of the percentage of BrdU positive nuclei in intestinal organoids from CTR and GCD–CeD. Columns represent the mean and bars the standard deviation. Student’s *t* test: * = *p* < 0.05.

**Figure 4 cells-12-00115-f004:**
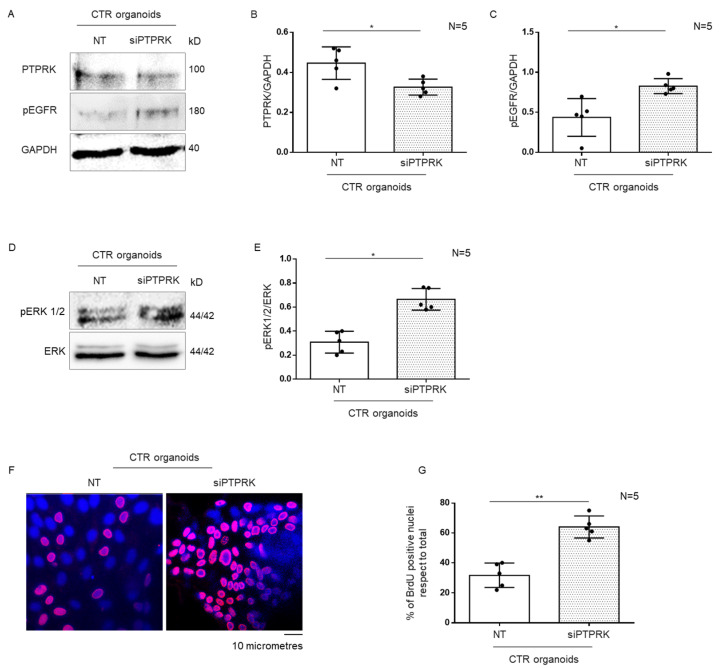
**Silencing PTPRK in CTR organoids induced increase inpEGFR, pERK and proliferation**. (**A**). Western blot analysis of total protein lysates of CTR organoids transfected and not with silencing PTPRK (siPTPRK). The membrane was blotted with antibodies against PTPRK, pEGFR and GAPDH as loading control. (**B**,**C**).Densitometric analysis of the PTPRK/GAPDH and pERK/GAPDH bands. Columns represent the mean and bars the standard deviation. Student’s *t* test: * = *p* < 0.05. (**D**). Upper lines were blotted with an antibody against pERK. Bottom lines were blotted with anti-ERK antibodies as loading control. (**E**). Densitometric analysis of the pERK/ERK bands. Columns represent the mean and bars the standard deviation. Student’s *t* test: * = *p* < 0.05. (**F**). Immunofluorescence images of anti-BrdU staining on intestinal organoids from CTR. Red nuclei are positive for BrdU staining. (**G**). Statistical analysis of the percentage of BrdU positive nuclei in intestinal organoids from CTR. Columns represent the mean and bars the standard deviation. Student’s *t* test: ** = *p* < 0.01.

**Figure 5 cells-12-00115-f005:**
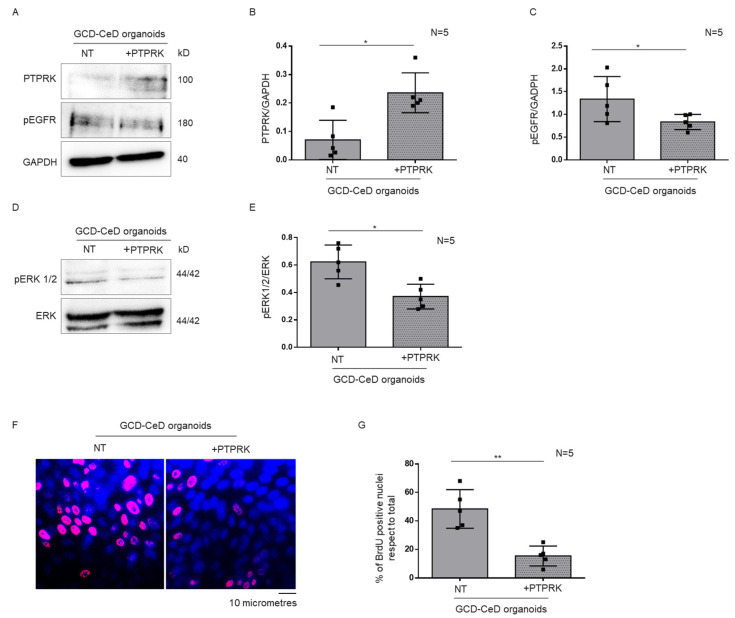
**Overexpression of PTPRK in CeD organoids induced a decrease inpEGFR, pERK and proliferation.** (**A**). Western blot analysis of total protein lysates of CeD organoids transfected and not with PTPRK. The membrane was blotted with antibodies against PTPRK, pEGFR and GAPDH as loading control. (**B**,**C**). Densitometric analysis of the PTPRK/GAPDH and pERK/GAPDH bands. Columns represent the mean and bars the standard deviation. Student’s *t* test: * = *p* < 0.05. (**D**). Upper lines were blotted with an antibody against pERK. Bottom lines were blotted with anti-ERK antibodies as loading control. (**E**). Densitometric analysis of the pERK/ERK bands. Columns represent the mean and bars the standard deviation. Student’s *t* test: * = *p* < 0.05. (**F**). Immunofluorescence images of intestinal organoids from GCD–CeD patients stained with anti-BrdU antibodies. Red nuclei are positive for BrdU staining. (**G**). Statistical analysis of the percentage of BrdU positive nuclei in intestinal organoids from CTR and GCD-CeD. Columns represent the mean and bars the standard deviation. Student’s *t* test: ** = *p* < 0.01.

**Table 1 cells-12-00115-t001:** Patients’ characteristics.

Patients	Range Age (Years)	Sex	Biopsy (Marsh Classification *)	Serum AntiTG2 (U/mL)	Anti-Endomysial Antibody (EMA)
Controls (N = 20)	9–17	M = 8 F = 12	20 = T0	0–1.7	Negative
GCD–CeD (N = 20)	4–16	M = 10 F = 10	6 = T3c 14 = T3 c/b	>50	Positive
POT–CeD (N = 8)	8–13	M = 5 F = 3	6 = T0 2 = T1	9–60	Positive

* T0: Normal; T1: infiltrative lesion; T3: Flat destructive lesion (b: moderate, c: total).

**Table 2 cells-12-00115-t002:** Organoids characteristics.

Patients	Range Age (Years)	Sex	Biopsy (Marsh Classification *)	Serum AntiTG2 (U/mL)	Anti-Endomysia Antibody (EMA)
Controls (N = 5)	4–15	M = 3 F = 2	5 = T0	0–1.9	Negative
GCD–CeD (N = 5)	8–17	M = 2 F = 3	3 = T3c 2 = T3 c/b	>50	Positive

* T0: Normal; T3: Flat destructive lesion (b: moderate, c: total).

**Table 3 cells-12-00115-t003:** Patients’ characteristics.

Patients	Range Age (Years)	Sex	Biopsy (Marsh Classification *)	Ki67 %
Controls (N = 4)	8–16	M = 2 F = 2	T0	22.5–33
GCD–CeD (N = 4)	3–15	M = 3 F = 1	T3c	52.7–62.2
POT–CeD (N = 4)	2–14	M = 1 F = 3	T0/T1	37–53

* T0: Normal; T1: infiltrative lesion; T3: Flat destructive lesion (c: total).

**Table 4 cells-12-00115-t004:** Quantitative values of an increase or a decrease in CeD patients’ biopsies.

CeD–Patients	PTPRK	pEGFR	pERK	Ki67 Positive Nuclei
12980 T3c	−2	+7.8	+2	+2.2
12985 T3c	−4.6	+9.3	+3.6	+2.3
12686 T3c	−2.3	+6.3	+3.5	+1.9
12994 T3c	−2	+7.1	+2.6	+2.2

CeD patients’ biopsies with low levels of PTPRK present high levels of EGFR and ERK phosphorylation together with increased Ki65 positive cells. The numbers represent quantitative values of an increase or a decrease of at least 2 times respect tothe mean of controls’ biopsies for the same parameter.

## Data Availability

Not applicable.

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
