# Peer review of "PTPRK, an EGFR Phosphatase, Is Decreased in CeD Biopsies and Intestinal Organoids"

_cells, 2022, doi:10.3390/cells12010115_

Round 1

Reviewer 1 Report

The study of Nanayakkara et al. addresses the mechanism of increased proliferation in celiac disease (CD) and focuses on the PTPRK, a phosphatase, which can activate EGFR. While PTPRK is known to be downregulated in CD, EGFR is upregulated as measured in CD biopsies by the Barone group. The experiments of the actual study are performed in CD (and sick control) biopsies as well as in organoids of CD and control patients. The analysis is done by immunofluorescence, Western blots, determination of mRNA levels by PCR analysis, proliferation assay by BrdU incorporation as well as silencing and overexpression of PTPRK in control/CD organoids. The experiments clearly show uniformly in CD biopsies as well as in CTR organoids that PTPRK regulates pEGFR, pERK and proliferation in CD. These results represent an important mosaic stone in the understanding of the complex pathogenesis of CD and should therefore be published.

Minor points to be addressed:

-       For a reader who is less familiar with abbreviations on CD it is hard to get into the issue when these are not explained consequently from the beginning of reading.

-       There are a lot of grammar mistakes, inconclusive sentences (especially in the methods part) like in 21, 28, 61-62, 132/3, 177/8, 185, 225, 242, 267-9,  

-       A more detailed/specified description of the results in the abstract will be more informative.

-       26: “… silencing an overexpression …“ instead of “…, si-RNA and overexpression …”

-       171: reference

-       182: how much protein was used for Wb?

-       The ki67 data should be shown either as mean ± standard deviation or median + range. The ki67 data represent fig 2F, so this should not be shown in the methods section, but rather be included in fig 2.

-       244: the authors should indicate all used siRNA including product numbers.

-       249: Did the authors incubate the cells for 72h without serum, or did they change the medium or add serum within the 72h?

-       261: Please indicate more detailed information about the transfection with respect to the overexpression of PTPRK

-       Can 2.4. and 2.10 be brought together?

-       Could the authors display actual values in table 4 – if at all – instead of semiquantitative values?

-       Do the described effects in organoids depend on the incubation or life span times of the organoids? At which time after seeding of the organoids were the analyses performed?  

-       489: Which cells in addition (to epithelial cells/enterocytes)?

-       506: … EGFR is delayed …

-       522: … overexpressing …

-       Can these results on PTPRK and EGFR discussed in relation to Notch and TGF-ß pathways and in the context of apoptosis (lymphoma genesis in CD) as a mechanism of villous atrophy?   

Reviewer 2 Report

please compare with other similar studies.

more quality of figure are requested
